# Spatial and Statistical Analysis of Operational Conditions Contributing to Marine Accidents in the Singapore Strait

Serdar Yildiz [1], Fatih Tonoğlu [2], Özkan Uğurlu [2,*], Sean Loughney [3] and Jin Wang [3,*]

1    World Maritime University, 211 18 Malmö, Sweden
2    Maritime Transportation and Management Engineering Department, Ordu University, Ordu 52400, Turkey
3    Liverpool Logistics, Offshore and Marine (LOOM) Research Institute, Faculty of Engineering and Technology, Liverpool John Moores University, Liverpool L3 3AF, UK
*    Correspondence: ougurlu@odu.edu.tr (Ö.U.); j.wang@ljmu.ac.uk (J.W.)

**Abstract:** Narrow waterways are important connection hubs, also known as logistics transfer nodes, within maritime transport, where maritime traffic can become very dense and congested. Heavy traffic, unsuitable environmental conditions and human errors make narrow waterways risky areas for marine accident occurrence. Accidents in narrow waterways cause ship damage, loss of cargo, loss of life and environmental disasters, as well as interruption of maritime transport and negative impact on the economy. Thus, the sustainability of navigational safety in narrow waterways has been the focus of attention of all beneficiaries in the maritime industry. The Singapore Strait is one of the busiest narrow waterways in the world in terms of the number of ships transiting. Sustaining and safe maritime transport in the Singapore Strait is significantly important for the sustainability of the global trade. Therefore, it is vitally important to appropriately identify the threats to safety of navigation in the Singapore Strait. In this study, the operational conditions that have played a role in the occurrence of accidents in the Singapore Strait are examined. For this purpose, using the Geographical Information System (GIS), the areas where marine accidents are concentrated were determined by the Kernel Density Analysis method and a "Marine Accidents Density Map" was created for the Singapore Strait. The relationship between the dense areas in the marine accidents density map and the operational conditions that play a role in the accidents in the Singapore Strait were examined using the Chi-Square Test and expert opinions. The results of the study indicate that if there is a condition (e.g., turning, joining to the traffic stream, or failure in propulsion/steering systems) that directly or indirectly disturbs the normal flow of traffic in the Singapore Strait, the risk of having an accident increases. The results of this study can be used to determine the measures to be taken for the prevention of possible accidents, as well as to help manage the risks associated with the ships that pass through the region.

**Keywords:** Singapore Strait; narrow waterways; marine accidents; Geographical Information Systems (GIS); spatial analysis

## 1. Introduction

National and international organisations publish many regulations to ensure safety in global maritime transport [1]. Despite the legal regulations and the measures taken, marine accidents remain a current threat to maritime safety [2,3]. In terms of world maritime shipping, 19,418 marine accidents occurred between 2014–2019 [4]. This situation reveals that accidents are a serious risk factor threatening maritime safety, as is frequently emphasised in literature studies [5,6]. The consequences of these accidents negatively affect the maritime sector in terms of property loss, economic loss, environmental damage and loss of life, as well as having a knock on effect to other industries [7–10]. One of the most recent examples is the accident in the Suez Canal in March 2021. The Suez Canal, which allows the passage of 1 million barrels of oil products per day, was closed to ship traffic for

6 days between 23 to 29 March 2021. The amount of compensation demanded from the shipowner company for the damage caused by accident is around 1 billion USD [11].

Narrow waterways are sea areas where the hazards faced by ships and the risks of accidents that may occur as a result of these hazards are the highest [12,13]. Compared to open waters, narrow waterways are sensitive areas in terms of marine accidents, where traffic density is higher, manoeuvring space is very limited, and ship speed is higher [14–16]. It is possible to divide the narrow waterways into two categories; canals and straits, which are important elements in connecting maritime trade. While canals are mostly man-made structures, constructed for purposes such as facilitating ship navigation, logistics, and agricultural activities, straits are natural formations. According to Britannica [17] there are over 100 navigable straits worldwide. As straits generally provide intercontinental connections and are transfer points used by many inland seas to access the open seas, maritime traffic is always very intense in these regions. In addition, since residential areas in the straits are very close to the shore, the consequences of any accident (such as loss of life, environmental pollution and ecosystem destruction) that may occur can be quite devastating [18–20]. Thus, this places further importance on navigational safety in these regions. The coastal state authorities establish a traffic flow pattern in the straits, and ships are expected to pass safely through the strait by complying with this traffic flow pattern as much as possible. Vessel Traffic Services (VTS), pilotage services and tugboat services were established in the straits in order to ensure the coordination and order of traffic during this passage.

The Singapore Strait is one of the most important straits that play an important role in maritime transport. The Singapore Strait is a 56.7 NM long, 16 NM wide between the Strait of Malacca in the west and the Strait of Karimata in the east. Traffic in the Singapore Strait is regulated by the Vessel Traffic Information System (VTIS) and is operated by the coastal state of Singapore. Approximately 70,000–80,000 ships pass through the strait each year, and it is named as the world's busiest narrow waterway in terms of tonnage [12,21]. In addition to the traffic density, the close proximity to islands and the shallowness of the strait further restrict navigation through the narrow waterway. The Singapore Strait is very narrow due to the islands in the region and the landmasses that have penetrated into the strait, for example the narrowest part is Tahong Buoy at 0.5 NM. There are anchorage areas at the east and west entrances of the Traffic Separation Scheme (TSS) in the strait and on the north side of the safe traffic lane. The northern lane of the traffic separation scheme is reserved as a single lane for east–west transit ships (average separation width 0.6 NM), whereas the south passage is a double lane for vessels transiting west to east (average separation width 0.8 NM) (Figure 1) [22].

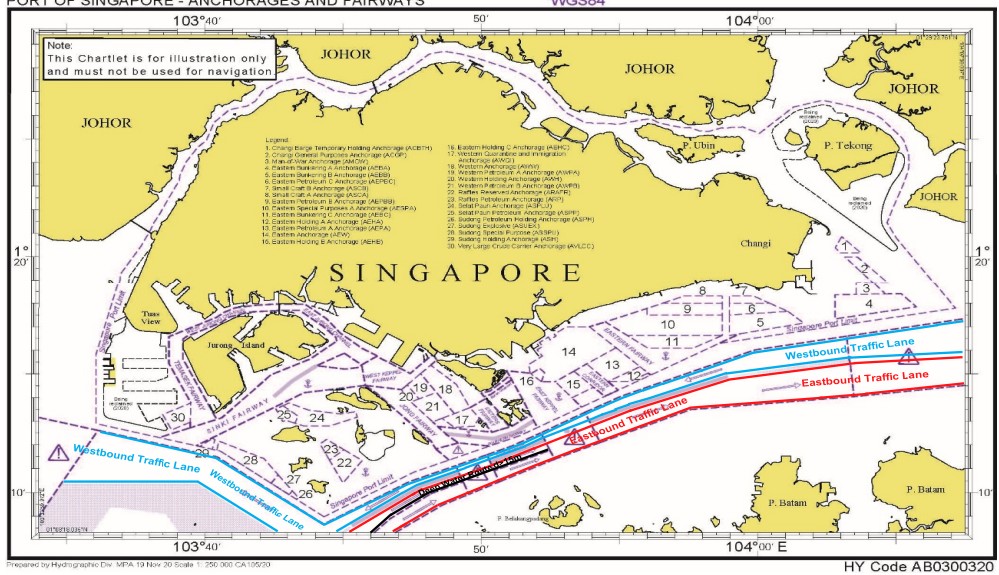

**Figure 1.** Distribution of Singapore Strait anchorages [22].

The Singapore Strait is among the riskiest and most dangerous narrow waterways in the world [23,24]. In this narrow waterway, accidents and the factors that cause them have a variable structure. Increasing ship traffic gradually makes this narrow waterway even more dangerous [18,19,25]. According to EMSA data, 230 marine accidents were reported in the Singapore Strait between 2011 and 2017 [26]. Numerical data and academic studies prove that the Singapore Strait is a high-risk navigational region. Research studies show that despite all of the precautions, accidents are still occurring [5,27,28]. On 13 January 2019, a collision occurred between a Hong Kong Flagged tanker and an Indonesian Flagged submarine pipeline vessel in the Singapore Strait. As a result of the accident, the pipeline vessel, which had a large hole on the port side, sank completely, and all its personnel were rescued with a search and rescue operation. There was no environmental pollution as a result of the accident, but the ship, worth 100 million USD, became completely unusable [29]. Similar marine accidents that have recently occurred in the Singapore Strait prove that the safety measures taken in narrow waterways should be constantly reviewed.

The most common accident types among marine accidents are collision-contact, sinking and grounding [7,30]. These marine accidents also occur frequently in narrow waterways [31]. All three accident types are closely related to narrow channel structure, traffic density and environmental conditions [32,33]. Operational conditions represent the final stage in the occurrence of a marine accident. It acts as a complementary element for unsafe acts that caused the accident. Unsafe acts are defined as actions taken by individuals or operators that directly lead to an accident. Some examples are errors, violations or abuses made by the operators. Every marine accident is associated with at least one operational condition, and they are divided into two subcategories: internal conditions and external conditions. Internal conditions include factors related to ship structure, and factors that affect ship movement which are partially controlled by operators. External conditions include natural conditions and non-ship factors that are not related to the ship's structure or not caused by human contribution and intervention. There is an interaction between operational conditions and unsafe acts rather than a cause-effect relationship and ship accidents tend to occur as a result of this interaction [24].

In this study, a marine accident density map was created with ArcMap 10.5 software (Esri, Redlands, CA, USA) using the marine accident data reported in the Singapore Strait. The relationship between operational conditions and accidents in areas where accidents are concentrated has been revealed. The study presents the marine accident density map for the Singapore Strait as well as the hazards arising from operational conditions in this narrow waterway. These hazards have been interpreted in line with the opinions of experts. The results of this study will contribute to raising awareness about the present hazards arising from operational conditions in the Singapore Strait.

## 2. Literature Review

There are many studies related to the effectiveness of maritime traffic and the sustainability of navigational safety in the Singapore Strait. Bateman et al. [12] determined the factors that threaten maritime safety based on the number of ships passing through the Straits of Malacca and Singapore. They focused on the types of ships, the commercial purposes of the countries in the region, and the factors that threaten maritime security (piracy, armed robbery, hijacking, etc.) and revealed their interrelationships. The authors emphasised that local traffic and fishing vessels pose a high risk regarding navigational safety in these straits.

Qu et al. [15] introduced three ship collision risk indices (index of speed dispersion, degree of acceleration/deceleration, and number of ship domain overlaps) for quantitative evaluation of ship collision risk in the Singapore Strait. Risk indices were created using real-time ship positions and navigation speeds. As a result of the study, the riskiest segments in the Singapore Strait were determined based on the estimation of these three risk indices. In addition, Qu et al. [15] revealed in their study that 25% of the ships navigating in the strait exceeded the speed limit, which increases the risk of collision. Indeed, higher speed

and/or speed variation makes it more difficult for navigators to react and respond based on common judgment when incidental scenarios occur. Speed changes in the Singapore Strait are more frequent when ships are about to cross, overtake, encounter or turn. Navigating above the permissible speed limit in the Singapore Strait will result in greater course changes between vessels at risk of collision. This will increase the probability of collision with other ships in an area where there is dense traffic.

In their study, Weng et al. [16] presented a probability equation for estimating collision frequency in the Singapore Strait using real-time ship motion data. As a result, they demonstrated that container ships have the highest frequency whereas Roll-On Roll-Off (Ro-Ro) and passenger ships have the lowest frequency. In addition, they determined the riskiest overtaking area, the riskiest bow crossing area, and the crossing area where collisions are most frequent. They found that westbound traffic in the Singapore Strait was riskier than eastbound traffic, and the frequency of collision was lower during the daytime than at night.

Meng et al. [34] analysed the characteristic of ship traffic by using instantaneous Automatic Identification System (AIS) data of 4 million ships passing through the Singapore Strait. As a result of the study, they stated that container ships (36.4%) have the largest share in the regional traffic, while Ro-Ro passenger ships (4.8%) have the lowest share. Furthermore, since tanker, bulk cargo, liquefied natural gas, and liquefied petroleum vessels are larger, faster, and have deeper draughts compared to other vessels, they stated that special attention should be paid to these vessels while arranging traffic in the Singapore Strait. They also revealed that the traffic in the VTIS East, in the Singapore Strait, is faster than the traffic in the VTIS West, and the area between longitudes 103°48′ E–104°05′ E has the highest traffic density.

Kang et al. [35] analysed traffic flow on fifteen route legs in the Singapore Strait, using more than 43 million AIS data points. As a result, they presented 75 diagrams based on four classical traffic flow models showing the theoretical capacity of each leg in the Singapore Strait. It was stated that the ship speed vs. traffic density relationship in the Singapore Strait is generally inversely proportional. It was emphasised that the ship speed vs. traffic density relationship has a complex structure which can vary irregularly depending on many factors such as officer on watch, master, a pilot being on board, weather conditions, ship size and ship type.

When the studies in the literature are examined, traffic density, efficiency modelling and estimation of the frequency of accidents in the Singapore Strait are frequently studied. It has been stated in previous studies that traffic density is variable and can be directly or indirectly affected by many environmental and operational factors. However, the number of studies focusing on operational conditions (environmental and ship-related) that play a complementary role in the occurrence of marine accidents, together with human factors, is quite limited. Therefore, it is important to determine these conditions comprehensively and to reveal their role in accidents to ensure the sustainability of maritime safety in the Singapore Strait. In this study, marine accidents in the Singapore Strait were analysed using Geographic Information Systems (GIS), Chi-Square independence test and expert opinions. The expert group in this study includes ship captains who have experience of sailing the Singapore Strait many times. This study aims to bridge the research gap and determine the operational conditions that play a role in the occurrence of accidents in the Singapore Strait.

## 3. Materials and Methods

This study examines marine accidents that occurred in the Singapore Strait during a period of 16 years between 2004–2019. Unlike marine accident analysis studies that are only made with theoretical applications and analytical evaluations, in this study, a theoretical-practical comparison was made by evaluating the results obtained from each stage of the study with an expert group consisting of maritime professionals who are experienced in narrow waterways. In this way, it is aimed to include risks that may be overlooked in the qualitative analysis of accident reports and analytical evaluations, but that exist in practice.

Thus, it is aimed to minimize the lack of data called "uncertainties" in safety assessment and risk analysis studies in the literature.

In the first step, accident reports in the Singapore Strait were collected, and an accident database of the region was created. Information on accidents (ship name, ship type, length, gross tonnage, accident date, etc.) has been obtained from a total of 17 databases (Table 1). This is to ensure that the scope of the study includes as many suitable accidents as possible. A total of 6548 accident data were found in the accident databases [36–41]. The spatial data of each accident was located on an electronic map, and was checked whether accident would be included in the study's data set. After the location analysis, the types of accidents were examined. Collision-contact, grounding and sinking accidents are within the scope of this study. As a result, 61 accidents are deemed suitable for this investigation although it would be more desirable if there could be more accidents of this kind considered. In all accidents in the data set, at least one of the ships involved in the accident is subject to IMO rules. The spatial analysis aims to identify the risky areas along the TSS in the Singapore Strait and to reveal the relationship between the accidents and the Operational Conditions in the risky areas. In order to conduct the spatial analysis, the positional data of the 61 accidents need to be converted into a format that ArcMap 10.5 software can process. Following the location processing, the spatial data of all accidents is mapped pointwise in ArcMap 10.5 software. In order to complete this, up-to-date raster navigational charts of the Singapore Strait, in vector format (two-dimensional) prepared with the World Geodetic System 1984 (WGS-84) datum, is required [42,43]. After obtaining the maps, spatial analysis is performed in three steps using ArcMap 10.5 software and IBM Statistical Package for the Social Sciences (SPSS) software [44,45]. In the first step, the point accident data is located on the nautical chart for the Singapore Strait. At the end of the first step, the distribution map of the accidents in the Singapore Strait is created.

**Table 1.** Marine accident databases.

| Country | Name | Abbreviation |
|---|---|---|
| Australia | Australian Transport Safety Bureau | ATSB |
| Canada | Transportation Safety Board of Canada | TSB |
| IMO | Global Integrated Shipping Information System | GISIS |
| Finland | Safety Investigation Authority | SIA |
| France | Civil Aviation Safety Invest. and Analysis Bureau | BEA |
| Europe | European Maritime Safety Agency | EMSA |
| Japan | Japan Transport Safety Board | JTSB |
| Netherlands | Dutch Safety Board | DSB |
| New Zealand | Transport Accident Investigation Commission | TAIC |
| Norway | Accident Investigation Board Norway | AIBN |
| Russia | Interstate Aviation Committee | IAC |
| Singapore | Air Accident Investigation Bureau of Singapore | AAIB |
| Sweden | Swedish Accident Investigation Authority | SAIA |
| China | Aviation Safety Council | ASC |
| Turkey | Transport Safety Investigation Center | UEIM |
| United Kingdom | Marine Accident Investigation Board | MAIB |
| United States | National Transportation Safety Board | NTSB |

In the second step, Hot Spot Analysis and Kernel Density Analysis methods are applied for the Singapore Strait using the geostatistical analysis tool of ArcMap 10.5 software. The Kernel Density Analysis method provides an estimation of the probability density function ($f(x)$) of any continuous random variable ($x$) by using non-parametric regression analysis. By using the sample data of an event or situation, it reveals the value range function of the probability of this event occurring in a certain neighbourhood. For example, let $x_1, x_2, x_3, \ldots, x_i$ be independently and identically distributed samples (a

number of accidents in a given area). The density distribution function $\hat{f}_h(x)$ of these samples is calculated using Equation (1) [46]:

$$\hat{f}_h(x) = \frac{1}{nh} \sum_{i=1}^{n} K\left(\frac{x - x_i}{h}\right) \tag{1}$$

where;

$\hat{f}_h(x)$: Kernel density distribution function

$K$: Kernel Function with symmetric probability density function and not a negative value

$h$: Correction parameter called search radius (bandwidth); h > 0 should always stand, but the dataset should be kept as small as it allows

$n$: Sample size

$x$: Kernel Centre (origin of the specified location for analysis)

$x_i$: $i$th sample

$x$-$x_i$: Distance between Kernel Centre and sample value (distance)

It has been observed that the hot spot analysis gives more consistent density analysis results in data that is concentrated at one point and does not show much distribution [47,48]. For this reason, it was decided that Kernel Density Analysis would be appropriate for data that is interrelated but also widely distributed, as well as clustering at fixed points such as marine accident data. Dense areas and kernel densities were determined by applying the Kernel Density Analysis method to point accident data in the Singapore Strait. The primary purpose of Kernel Density Analysis is to generate density distribution maps at the desired search radius from the kernel points where the accidents occur [47]. The search radius is required to calculate Kernel densities in the geographic area where the accidents (point data) are located. The Kernel density value is highest at the centre of the accident and decreases with distance, so that the Kernel density value reaches zero at the far end of the search radius. The optimum selection of the kernel search radius is crucial for the accurate detection of dense accident areas [49]. If the search radius is defined as too high, non-dense areas will also come out as "high density". If the search radius is defined as too low, hot spots will be detected instead of dense areas. Both choices will produce erroneous outputs, which in turn will lead to erroneous results. In this study, while applying Kernel Density Analysis, the bandwidth and kernel radius were optimised by considering the approaches applied in studies in the literature [50–52]. Trials were made for the Singapore Strait in the ranges of ($0.7° \times 0.7°$), ($0.5° \times 0.5°$), ($0.3° \times 0.3°$), ($0.1° \times 0.1°$), ($0.09° \times 0.09°$), ($0.07° \times 0.07°$), ($0.05° \times 0.05°$), ($0.03° \times 0.03°$), ($0.01° \times 0.01°$). The optimum Kernel bandwidth was determined as $0.05° \times 0.05°$, considering the geographical structure of the Singapore Strait, the arrangement of anchorage areas, the arrangement of the traffic separation scheme and the point (spatial) distribution of the accidents [46]. As a result of the application, the "Marine Accidents Density Map" was obtained for accidents in the Singapore Strait.

The last step of the spatial analysis aims to determine the relationship between the Operational Conditions affecting the accidents in the sea areas where accidents are concentrated based on accident type and severity. The Chi-square independence test is one of the methods used in the literature to compare observed results with expected results [53–55]. There are three commonly used Chi-Square tests, these are the goodness of fit test, homogeneity test and independence test, where the purpose of each test is different. A good fit test is used to test the suitability of the sample to a particular data set or distribution (Binomial, Poisson, Discrete, Normal). The homogeneity test is used to measure whether a sample of a selected volume of sample selected from the population varies in similar characteristics to the population [46]. The Chi-square independence test is used to determine whether there is a statistically significant relationship between two variables [53,56]. One major advantage of the Chi-Square independence test is that it can be applied to nominal data as well as numerical data [57–59]. Therefore, it has been decided to use the Chi-Square independence test since a study will be conducted to examine the relationship between the variables.

The existence of a significant relationship between marine accident types (collision-contact, grounding, sinking) and Operational Conditions (ship age, ship length, ship type, day status (night-day), etc.) in very high density and high density accident areas in the Singapore Strait is analysed. Then, the existence of a significant relationship between the accident severity (very serious, serious, less serious) of the marine accidents that occurred in these sea areas and the Operational Conditions is examined. According to IMO's "Casualty Investigation Code" in its updated version (IMO Circular MSC-MEPC.3/Circ.3), very serious casualties mean a marine casualty involving: the total loss of the ship or; a death or; severe damage to the environment. Serious casualties are casualties to ships which do not qualify as very serious casualties and which involve a fire, explosion, collision, grounding, contact, heavy weather damage, ice damage, hull cracking, or suspected hull defect, etc., resulting in: immobilization of main engines, extensive accommodation damage, severe structural damage, such as penetration of the hull under water, etc., rendering the ship unfit to proceed, or pollution (regardless of quantity); and/or a breakdown necessitating towage or shore assistance. Less serious casualties are casualties which do not qualify as very serious casualties or serious casualties.

Finally, the existence of a significant relationship between the Kernel Density (very high, high) of the area where these marine accidents occurred, and the Operational Conditions is evaluated. For the purpose of the study, 18 null hypotheses (H0–H17) for which the Chi-Square independence test will be applied have been established (Table 2). IBM SPSS 25.0 software was used to implement the Chi-Square independence test [45]. At the end of this step, the relationship of the accidents that occurred in the Singapore Strait with the Operational Conditions is clearly demonstrated. Chi-square test results and spatial analysis results obtained at the end of the study were shared with the maritime expert group. The expert group consists of 13 people, each of whom has transited through the Singapore Strait multiple times, holds the competency of oceangoing master, and are familiar with the geographical area and the risks related to safe navigation. Experts were asked to evaluate and interpret the impact of each operational condition (identified in the study) on the accidents. At this stage, existing risks in marine areas were determined with the participation of experienced maritime professionals. Information about the experts whose opinions were consulted in the study is given in Table 3.

**Table 2.** Chi-square hypotheses established in the study.

| Hypothesis | |
| --- | --- |
| There is no significant relationship between accident type and | H00: age of ship.<br>H01: ship length.<br>H02: type of ship.<br>H03: accident severity.<br>H04: season.<br>H05: day status (day/night).<br>H06: Kernel Density. |
| There is no significant relationship between accident severity and | H07: age of ship.<br>H08: ship length.<br>H09: type of ship.<br>H10: season.<br>H11: day status (day/night).<br>H12: Kernel Density. |
| There is no significant relationship between Kernel Density and | H13: age of ship.<br>H14: ship length.<br>H15: type of ship.<br>H16: season.<br>H17: day status (day/night). |

**Table 3.** Experts whose opinions were taken at the end of the spatial analysis and their characteristics.

| No. | Current Position | Experience In Current Rank (Years) | Total (Years) | Previous Sea Service | | Number Of Passages In Singapore Strait |
| | | | | Last Competency | Experience Master (Month) | |
|---|---|---|---|---|---|---|
| 1 | Faculty member in maritime university | 8 | 15 | Oceangoing master | 4 | >50 |
| 2 | Oceangoing master | 5 | 12 | Oceangoing master | 70 | >20 |
| 3 | Vessel Traffic Service Operator | 3 | 13 | Oceangoing master | 24 | >5 |
| 4 | Faculty member in maritime university | 4 | 11 | Oceangoing master | 48 | >20 |
| 5 | Oceangoing master | 8 | 14 | Oceangoing master | 96 | >20 |
| 6 | Oceangoing master | 6 | 10 | Oceangoing master | 72 | >10 |
| 7 | Chief officer of coastal safety tug | 4 | 8 | Oceangoing master | 6 | >15 |
| 8 | Vessel Traffic Service Operator | 3 | 5 | Chief Oceangoing Officer | - | >20 |
| 9 | Maritime pilot | 3 | 12 | Oceangoing master | 48 | >20 |
| 10 | Vessel Traffic Service Operator | 6 | 12 | Oceangoing master | 72 | >15 |
| 11 | Maritime pilot | 2 | 14 | Oceangoing master | 72 | >100 |
| 12 | Maritime pilot | 16 | 20 | Oceangoing master | 108 | >20 |
| 13 | Officials of Istanbul Technical University Turkish Straits Maritime Application and Research Centre | 30 | 30 | Oceangoing master | 120 | >20 |

## 4. Results and Discussion

Spatial analysis, Kernel Density Analysis and Chi-Square independence test results are presented in this section. The relationship between accident occurrence in the Singapore Strait and the operational conditions has been revealed based on expert opinions.

When the spatial distribution of the accidents occurring in the Singapore Strait is examined, it can be seen that the accidents are concentrated in the narrowest parts of the passage (Figure 2). In addition, accidents also frequently occur in the anchorage area of the strait at the Malacca Strait side and at the southern entrance of the strait. The most common accident type in the analysed accident data set is collision (90.2%). Accidents were more intense in monsoon periods, December to early March-Northeast Monsoon (24.6%), and June to September-Southwest Monsoon (42.6%) [60]. In parallel with the distribution of ships passing through the Singapore Strait according to the ship type; dry cargo (32.8%), container (29.5%) and tanker (27.9%) ships are the types with the most frequent accidents, respectively. In addition, due to its negative effect on proper lookout, it was observed that if the daylight is low, the accidents occur 2.4 times more frequently than during the daytime, during twilight and dark hours (Table 4). The accident-prone areas in the Singapore Strait are divided into 5 density categories (Very high (VH), High (H), Medium (M), Low (L), and Very low (VL)) based on Kernel density values. The Kernel Density Map generated as a result of the application is presented in Figure 3. Within the TSS, 7 "very high density" sea areas (28 accidents) and 11 "high density" sea areas (20 accidents) were identified. In total, 78.7% of the accidents in the Singapore Strait occurred in sea areas in these two density categories.

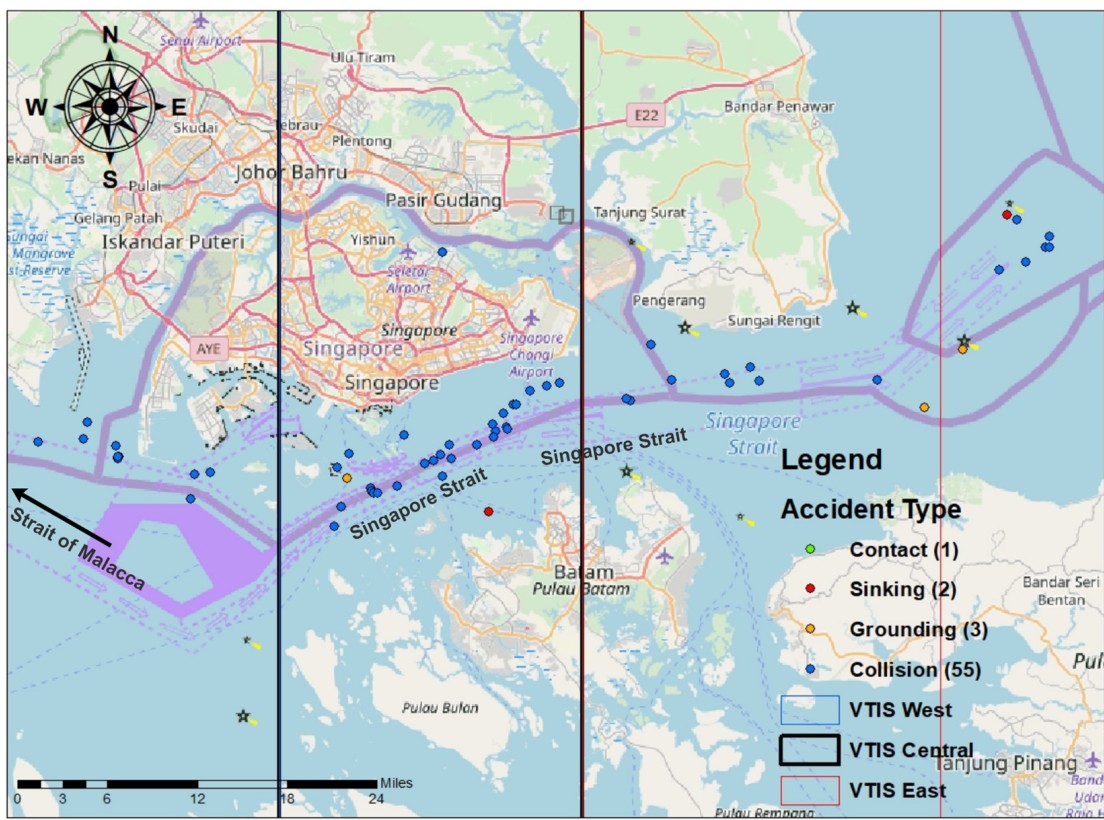

**Figure 2.** Point distribution of accidents in the Singapore Strait.

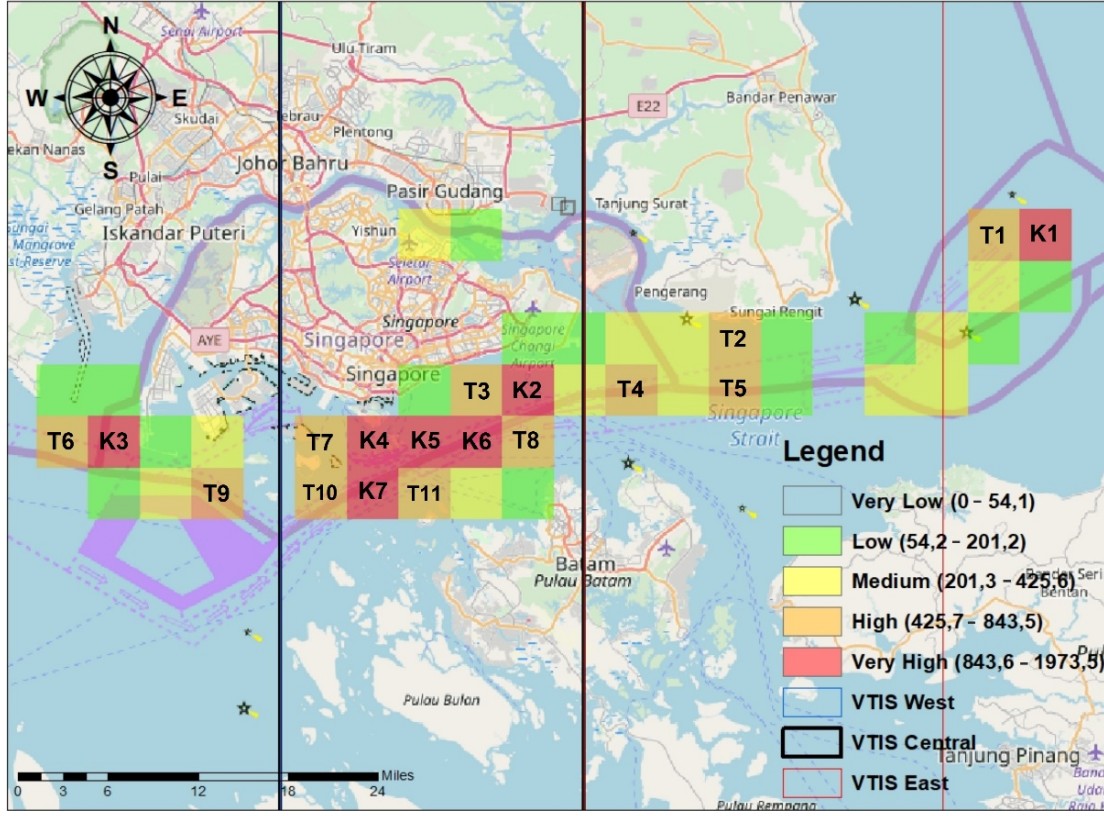

**Figure 3.** Singapore Strait Kernel density map.

**Table 4.** Distribution of frequency of accidents in the Singapore Strait by operational conditions.

| Operational Conditions | | Singapore Strait (N = 61) | | Singapore Strait (VH + H = 48) ƒ | | % | |
|---|---|---|---|---|---|---|---|
| | | ƒ | % | VH | H | VH | H |
| Ship Type | Dry Cargo | 20 | 32.8 | 10 | 7 | 35.7 | 35 |
| | Tanker | 17 | 27.9 | 7 | 6 | 25 | 30 |
| | Container Ship | 18 | 29.5 | 10 | 6 | 35.7 | 30 |
| | Other (RoRo, Passenger, etc.) | 6 | 9.84 | 1 | 1 | 3.6 | 5 |
| Ship Length | Length Overall (LOA) ≤ 100 | 3 | 4.92 | 0 | 2 | 0 | 10 |
| | 100 < LOA ≤ 150 | 3 | 4.92 | 1 | 0 | 3.6 | 0 |
| | 150 < LOA | 55 | 90.2 | 27 | 18 | 96.4 | 90 |
| Ship Age | Age ≤ 10 | 32 | 52.5 | 15 | 12 | 53.6 | 60 |
| | 10 < Age ≤ 30 | 26 | 42.6 | 12 | 7 | 42.9 | 35 |
| | 30 < Age | 3 | 4.92 | 1 | 1 | 3.6 | 5 |
| Season | Spring | 12 | 19.7 | 9 | 1 | 32.1 | 5 |
| | Summer | 26 | 42.6 | 9 | 10 | 32.1 | 50 |
| | Autumn | 8 | 13.1 | 3 | 3 | 10.7 | 15 |
| | Winter | 15 | 24.6 | 7 | 6 | 25 | 30 |
| Status of the Day | Day (06:01–18:00) | 18 | 29.5 | 5 | 7 | 17.9 | 35 |
| | Night (18:01–06:00) | 43 | 70.5 | 23 | 13 | 82.1 | 65 |
| Accident Type | Grounding | 3 | 4.92 | 1 | 1 | 3.6 | 5 |
| | Contact | 1 | 1.6 | 0 | 0 | 0 | 0 |
| | Collision | 55 | 90.2 | 27 | 19 | 96.4 | 95 |
| | Sinking | 2 | 3.28 | 0 | 0 | 0 | 0 |
| Accident Severity | Less Serious | 11 | 18 | 4 | 5 | 14.3 | 25 |
| | Serious | 38 | 62.3 | 18 | 13 | 64.3 | 65 |
| | Very Serious | 12 | 19.7 | 6 | 2 | 21.4 | 10 |

Kernel density analysis indicated that "very high density" sea areas in the Singapore Strait are concentrated in the VTIS Central sector, where the TSS is at its narrowest, and anchorage areas are concentrated. "High density" sea areas, on the other hand, are more prevalent in the VTIS Central but are distributed across all sectors in the Singapore Strait (Figure 3).

Sea areas with "Very high density" in the Singapore Strait include the northern approach close to the VTIS East sector (K1), the narrowest part of the strait (K5, K6, K7) in the VTIS Central sector, and the areas where anchorage areas concentrated in the sides of the strait (K2 and K4). In addition to these areas, the K3 area, located at the west entrance of Johor Strait within the VTIS West sector, is one of the "very high density" areas. On the other hand, no "very high density" sea areas were identified in the VTIS East sector, where the safe waterway is relatively wider, and there are no anchorage areas (Figure 3). When the "high density" areas are examined, the T1 area located at the eastern approach of the strait close to the VTIS East sector and the T2, T4 and T5 located within the VTIS East sector are high density areas. In the VTIS Central sector, T3, T7, T8, T10 and T11 are "high density" areas. In addition, T6 and T9, located on the western approach of the strait within the VTIS West sector, were identified as "high density" areas (Figure 3). T1, T2, T4, T5 and T6 are the areas where the waterway (navigation fairway) is relatively wider. Therefore, the most accident-prone VTIS sector in the Singapore Strait is the Central sector. The Kernel density map generated in this study supports the results presented in the study of Qu et al. [15] that the highest risk in Singapore Strait Traffic Separation Scheme is in the route legs in the VTIS Central sector.

The most common types of accidents in "very high density" areas are collision (96.4%) and groundings (3.6%), respectively. Similarly, the most frequent accident types in "high density" areas are collision (95.0%) and grounding (5.0%), respectively (Table 4). There were no sinking and contact accidents that occurred in those very high and high density

sea areas. Kang et al. [35] stated that the ships had to slow down when approaching the areas where the separation scheme is narrowed down and where there are sharp turns (waypoints). Meng [34] analysed the AIS data of ships passing through the Singapore Strait and revealed that westbound traffic is more congested than eastbound traffic in the area where the separation line (fairway) reaches its narrowest point. The fact that the westbound separation line is much narrower means that traffic congestion increases and forces the ships to slow down. As a result, there is an increase in overtaking situations [16]. It is known that the risks of collisions increase in restricted waters with ships sailing at different speeds [61]. Therefore, the results of this study are compatible with the studies in the literature.

When the operational conditions in accidents occurring in "very high density" areas are examined, the most common ships are longer than 150 m (96.4%) in length, ships with an age of 10 years (53.6%) and younger, ships carrying dry cargo (35.7%) and container (35.7%). It is understood that accidents in "very high density" areas occur mostly at night (82.1%) and increase during spring (32.1%) and summer (32.1%) seasons. In the "high density" regions, dry cargo ships (35%) were the ship type most frequently involved in the accident. Similar to "very high density" areas, most of the vessels involved in the accident were under 10 years of age (60%) and over 150 m (90%) in length. Accidents in "high density" areas occurred mostly during the summer season (50%) and at night (65%) (Table 4). Since the Singapore Strait is located close to the equatorial (tropical) belt, the seasonal distribution of accidents is important for understanding the effects of weather and sea conditions. Singapore has two main monsoon seasons: the Northeast Monsoon Season (December–March) and the Southwest Monsoon Season (June–September). During the Southwest Monsoon Season (June–September), the occasional "Sumatra Storms" with gusts of 40–80 km/h can occur between predawn hours and noon and affect navigation in the Singapore Strait [60]. Tropical storms are seen in the region during the spring and summer seasons hence, these seasons can result in increased accident occurrence. These results are rational due to the seasonal transitions in the tropical zone and the tropical storms being active in the summer months. These results support studies that show a close relationship between weather/sea conditions and accidents [62–64]. The fact that the accidents in the data set are concentrated at night shows that the accident frequencies may be correlated to the time of passage (status of the day).

Contact and sinking accidents were not observed in the "very high density" and "high density" areas in the Singapore Strait. For this reason, while applying the Chi-square tests, two categorical variables were considered under the accident type: collision and grounding (N = 48 accidents). According to the Chi-square test results, a significant relationship ($p < 0.05$) was found between the type of accident and season and the type of accident and status of the day. However, there is no statistically significant relationship identified between "the accident severity", "the density of the Kernel areas where the accidents occurred", and the "operational conditions" (Table 5).

When the cross table between the accident type and the season in which the accident occurred is examined, it is seen that the collisions are concentrated during the summer season (41.3%). Following summer, winter (28.3%) and spring (21.7%) are the seasons in which collisions frequently occur. On the other hand, groundings in "very high density" and "high density" areas occurred more frequently during the autumn, unlike collisions (Table 6). When the relationship between collision accidents and the status of the day (daylight) is examined, it is seen that night passages are 3 times more accident-prone (78.3%). This result supports the studies in the literature stating that day/night conditions play an active role in accidents in restricted waterways [18,61]. Similarly, experts in this study emphasised that night passages through the Singapore Strait are riskier in terms of accidents as opposed to passages during the daytime.

**Table 5.** Chi-square test results of Singapore Strait.

| | Pairwise Comparisons (Test Hypotheses) | Singapore Strait Significant Relationship | Significance (*P*) |
|---|---|---|---|
| Accident Type | Ship Age | No | 0.444 |
| | Ship Length | No | 0.933 |
| | Ship Type | No | 0.704 |
| | Accident Severity | No | 0.742 |
| | Season | Yes | 0.002 |
| | Status of the Day | Yes | 0.012 |
| | Density of Kernel Area | No | 0.807 |
| Accident Severity | Ship Age | No | 0.499 |
| | Ship Length | No | 0.781 |
| | Ship Type | No | 0.917 |
| | Season | No | 0.457 |
| | Status of the Day | No | 0.276 |
| | Density of Kernel Area | No | 0.443 |
| Density of Kernel Area | Ship Age | No | 0.850 |
| | Ship Length | No | 0.168 |
| | Ship Type | No | 0.964 |
| | Season | No | 0.148 |
| | Status of the Day | No | 0.176 |

**Table 6.** Cross-table of accident type-season and accident type- status of the day for Singapore Strait.

| | | | Season | | | | Status Of The Day | |
|---|---|---|---|---|---|---|---|---|
| | | | Spring | Summer | Autumn | Winter | Daytime | Night |
| Accident Type | Grounding | Number | 0 | 0 | 2 | 0 | 2 | 0 |
| | | Accident Type (%) | 0.0 | 0.0 | 100.0 | 0.0 | 100.0 | 0.0 |
| | | Season-Status of the day (%) | 0.0 | 0.0 | 33.3 | 0.0 | 16.7 | 0.0 |
| | Collision | Number | 10 | 19 | 4 | 13 | 10 | 36 |
| | | Accident Type (%) | 21.7 | 41.3 | 8.7 | 28.3 | 21.7 | 78.3 |
| | | Season-Status of the day (%) | 100.0 | 100.0 | 66.7 | 100.0 | 83.3 | 100.0 |
| | Total | Number | 10 | 19 | 6 | 13 | 12 | 36 |
| | | Accident Type (%) | 20.8 | 39.6 | 12.5 | 27.1 | 25.0 | 75.0 |

Spatial analysis and Chi-square results were presented to the expert group in the study. Based on the comments of the experts, the operational conditions affecting the accidents in the Singapore Strait are presented in Table 7. The table shows how many of the interviewed experts considered the relevant operational condition as a contributing factor to accidents. All of the experts participating in the study stated that the status of the day (daytime/night) during the passage, local traffic, and the number of previous transit experiences of the ship's master are the operational conditions that must be considered during passage planning. In addition, more than 80% of the experts emphasised that the narrowest part of the strait (where the fairway is narrow), high intensity shore lighting, sharp turning points, current speed, current direction and high ship density in the anchorage areas are the main factors that increase the risk of accidents. They pointed out that the age of the ship alone does not increase the risk of accident, and that this operational condition should be evaluated together with the planned maintenance system and its enforcement on the ship. Instead of focusing on the season as an operational condition, experts suggested that weather and sea conditions should be evaluated separately, and monsoon periods, tropical storms, and sudden weather changes should be also considered (Table 7).

**Table 7.** Expert judgments on the results of the study and operational conditions.

| Operational Conditions | Singapore Strait (N = 13 Experts) | | | | | |
| | Effective | | Not Sure | | Not Effective | |
| | *f* | % | *f* | % | *f* | % |
| --- | --- | --- | --- | --- | --- | --- |
| Ship age | 4 | 31% | 4 | 31% | 5 | 38% |
| Ship length | 10 | 77% | 1 | 8% | 2 | 15% |
| Ship type | 10 | 77% | 0 | - | 3 | 23% |
| Season | 3 | 23% | 0 | - | 10 | 77% |
| Status of the day | 13 | 100% | 0 | - | 0 | - |
| Ship speed | 10 | 77% | 0 | - | 3 | 23% |
| Ship draught | 10 | 77% | 0 | - | 3 | 23% |
| Wind direction | 8 | 61% | 1 | 8% | 4 | 31% |
| Wind speed | 8 | 61% | 1 | 8% | 4 | 31% |
| Current direction | 11 | 84% | 1 | 8% | 1 | 8% |
| Current speed | 11 | 84% | 1 | 8% | 1 | 8% |
| Master's number of passages | 13 | 100% | 0 | - | 0 | - |
| Narrowest part of the strait | 12 | 92% | 0 | - | 1 | 8% |
| Sharpest turn along the strait | 11 | 84% | 1 | 8% | 1 | 8% |
| Local traffic | 13 | 100% | 0 | - | 0 | - |
| Intensity of shore lighting | 12 | 92% | 0 | - | 1 | 8% |
| Capacity and occupancy rate of anchorage areas | 11 | 84% | 1 | 8% | 1 | 8% |

## 5. Conclusions

One of the most effective methods adopted for the prevention of marine accidents is the planning and management of future operations with the lessons learned from previous accidents/unsafe acts. It can be said that most of the rules and regulations in the maritime industry have emerged in this way. To maintain safe maritime transport in the Singapore Strait is extremely important for global trade. As a result, it is vitally important to appropriately identify the threats to safe navigation in the Singapore Strait. This study examines the relationship between accidents and operational conditions in the Singapore Strait, one of the world's busiest narrow waterways. For this purpose, using the Geographical Information System (GIS), the areas where maritime accidents are concentrated were determined by the Kernel Density Analysis method and a "Marine Accidents Density Map" was created for the Singapore Strait. The relationship between the dense areas on the "Marine Accidents Density Map" and the Operational Conditions that played a role in the accidents in the Singapore Strait was evaluated based on the Chi-Square Test and expert opinions. According to the Kernel Density Map, which shows the areas where the accidents have occurred frequently in the Singapore Strait, the shallow areas, the areas where the safe waterway is narrow and the areas where there are entrances and exits to the TSS are the riskiest. In other words, no matter how dense the current traffic is, it does not pose a significant risk if it can continue in its usual flow in a planned and organised manner. However, if there is a condition that directly or indirectly affects/disturbs the usual flow of the traffic (turning, joining, leaving, malfunctioning in propulsion and steering systems, etc.), the risk of accident occurrence increases. The results of this study highlight the points that need to be emphasized in the measures that can be taken to prevent similar accidents in the future.

The "high density" and "very high density" areas in the "Marine Accidents Density Map" for the Singapore Strait indicate where operational conditions more likely lead to accidents along the strait. The key advantages and purpose of the production of the Kernel density map reside in the visualisation of the risk, and the ease of interpretation and understanding by possible users (VTS operators, pilots, masters and officers onboard). In addition, the map can be easily reconfigured in line with the user's request, considering various attributes such as ship type, ship length, the severity of consequences, etc. The accident-prone areas on the map can also be easily updated by updating the study's accident database in accordance with user preferences. The methodology applied in this

study to generate the Singapore Strait Kernel density map can also be applied to narrow waterways in other geographical locations. These maps can be used by ship masters, pilots, and vessel traffic operators to increase situational awareness, familiarity with the area, and traffic characteristics. Future risk analysis and traffic modelling studies may provide more insights in accident occurrence if the operational conditions, presented as a result of this study, are taken into account as dependent variables (parameters).

**Author Contributions:** Conceptualization, Ö.U. and S.Y.; methodology, S.Y., F.T. and Ö.U.; software, S.Y.; validation, S.Y. and F.T.; formal analysis, S.Y.; investigation, S.Y.; resources, F.T.; data curation, S.Y. and F.T.; writing—original draft preparation, S.Y., F.T. and Ö.U.; writing—review and editing, Ö.U., S.L. and J.W.; visualization, S.Y.; supervision, J.W. and Ö.U.; project administration, Ö.U., S.L. and J.W.; funding acquisition, Ö.U., S.L. and J.W. All authors have read and agreed to the published version of the manuscript.

**Funding:** "This research was funded by International Association of Maritime Universities (IAMU), grant number FY 2020-Young Academic Staff". The project name is "Maritime Risk Evaluation and Safety Optimization in Narrow Straits: A Case Study in Istanbul Strait and English Channel (M-REASONS)".

**Institutional Review Board Statement:** Not applicable.

**Informed Consent Statement:** Not applicable.

**Data Availability Statement:** Not applicable.

**Conflicts of Interest:** The authors have no conflict of interest and unanimously agree to submit the manuscript to the journal.

## Nomenclature

| | | | |
|---|---|---|---|
| AIS | Automatic Identification System | Ro-Ro | Roll-on Roll-off |
| EMSA | European Maritime Safety Agency | SPSS | Statistical Package for the Social Sciences |
| GIS | Geographical Information System | TSS | Traffic Separation Scheme |
| IMO | International Maritime Organization | VTIS | Vessel Traffic Information System |
| LOA | Length Overall | VTS | Vessel Traffic Services |
| NM | Nautical Miles | WGS-84 | World Geodetic System 1984 |

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
