# Peer review of "Spatial and Statistical Analysis of Operational Conditions Contributing to Marine Accidents in the Singapore Strait"

_jmse, doi:10.3390/jmse10122001_

Round 1
Reviewer 1 Report
Please refer to the attachment

Author Response
We would like to thank you for your interest in our paper. We think that our article is clearer and more understandable with the changes we have made in light of your comments and suggestions. Responses to your comments are in the attached file.

Reviewer 2 Report
The manuscript describes the analysis of traffic safety in a high density area, Singapore Strait. This is done by collecting documented accidents, and marking them for location, type of ship, time of day, and other relevant factors.
The problem is clear, the structure of the manuscript well laid out, the data seems to be correct, the techniques used logical, the conclusions are supported by the analysis.
In general, the study is acceptable.
It does however have some shortcomings that need to be resolved. I will discuss them in order of the text, and some minor issues afterwards.
At the end of the introduction it needs a clear paragraph with what question will be answered in the study.
The analysis technique is using a kernel to describe the distribution, but the kernel size and shape are never discussed. This is an omission.
The authors use some commercial software, but do not describe adequately what the different techniques provided by the software actually calculate. The results must be independent of the software package used. A reader must be able to reproduce the results with any general-purpose or statistical software as well. Please explain in more detail what is calculated.
In table 4 accident frequencies are provided, but base lines are omitted: if container ships are 80% of all ships but are involved in 29.5% of the accidents they are doing pretty well. This might have a serious effect on the analysis.
That same table (4) has additional columns that need clarification.
There is a set of hypothesis based on objective criteria, and there are "experts" who evaluate each of the accidents included in the study. The objective criteria do not match the subjective qualifications. It seems to me that the subjective qualification better describe relevant factors, and the can easily be transformed into objective criteria. Why is this not done?
As a result of the points above, the conclusions are not very strong.
Some minor points:
The first author works in Sweden, but no literature describing traffic safety in the vicinity of Sweden is used. I wonder why.
line 29: joining the traffic stream
line 124: why does exceeding the speed limit increase risk? (this in not self explaining).
line 146: ship speed vs. traffic density
line 167: "who are experienced in narrow waterways": rephrase
line 179: make the selection positive
table 2: rephrase, removing most identical words
table 3, no 13: this is only one ex-captain
table 4: ship size: odd categories
table 4: accident severity: very subjective
Author Response
Thank you for your interest in our paper and for giving you the opportunity to revise it. Your comments and suggestions have helped us improve the manuscript. We have made the necessary revisions to the paper in response to your recommendations. Following the revisions, we believe that the paper has become clearer. Responses to your comments file is attached.

Round 2
Reviewer 1 Report
NIL
Author Response
Dear Reviewer,
We appreciate your valuable and constructive comments pertaining to our paper and have endeavoured to revise it accordingly. We are of the opinion that our article is clearer and more understandable with the changes we have made in the light of your comments and suggestions.
Kind regards.
Reviewer 2 Report
There is one point I would like to make: In Europe SS stands for Schutzstaffel, a criminal organisation of the German Nazi's, convicted for war crimes and crimes against humanity. I advice not to abbreviate Singapore Strait to SS. I know steam ships are similarly abbreviated, but this predates the war, and is always followed by the ship's name.
Author Response
Dear Reviewer,
Thank you for your interest in our article. We are of the opinion that our article is clearer and more understandable with the changes we have made in the light of your comments and suggestions.
Reviewer 2 comment 1:
There is one point I would like to make: In Europe SS stands for Schutzstaffel, a criminal organisation of the German Nazi's, convicted for war crimes and crimes against humanity. I advice not to abbreviate Singapore Strait to SS. I know steam ships are similarly abbreviated, but this predates the war, and is always followed by the ship's name.
Response to Reviewer 2 comment 1:
Based on your comment, all SS abbreviations in the article have been revised to Singapore Strait.